# Learning within Sleeping: A Brain-Inspired Bayesian Continual Learning Framework

## Abstract

Bayesian-based methods have emerged as an effective approach in continual learning (CL) to solve catastrophic forgetting. One prominent example is Variational Continual Learning (VCL), which demonstrates remarkable performance in task-incremental learning (task-IL). However, class-incremental learning (class-IL) is still challenging for the VCL, and the reasons behind this limitation remain unclear. Relying on the sophisticated neural mechanisms, particularly the mechanism of memory consolidation during sleep, the human brain possesses inherent advantages for both task-IL and class-IL scenarios, which provides insight for a brain-inspired VCL. To identify the reasons for the inadequacy of VCL in class-IL, we first conduct a comprehensive theoretical analysis of VCL. On this basis, we propose a novel bayesian framework named as Learning within Sleeping (LwS) by leveraging the memory consolidation. By simulating the distribution integration and generalization observed during memory consolidation in sleep, LwS achieves the idea of prior knowledge guiding posterior knowledge learning as in VCL. In addition, with emulating the process of memory reactivation of the brain, LwS imposes a constraint on feature invariance to mitigate forgetting learned knowledge. Experimental results demonstrate that LwS outperforms both Bayesian and non-Bayesian methods in task-IL and class-IL scenarios, which further indicates the effectiveness of incorporating brain mechanisms on designing novel approaches for CL.

## 1 Introduction

Continual Learning (CL) aims to train neural networks on sequential tasks while retaining memories acquired from past tasks within limited resources (Wang et al. (2022)). One significant challenge in CL is the occurrence of *catastrophic forgetting* (Kim et al. (2022)), which refers to the phenomenon where training a model with new information disrupts previously learned knowledge (Parisi et al. (2019)). In recent years, there has been a growing body of research focusing on the application of Bayesian inference frameworks in CL, which is well-suited due to their larger parameter space and recursive nature. One notable approach is the online variational inference (VI) method combined with Monte Carlo VI, as demonstrated in VCL (Nguyen et al. (2018)). VCL effectively combines the previous posterior distribution from old tasks as the prior distribution for the current task, along with the likelihood from the current task serving as the new posterior distribution. Building upon VCL, subsequent studies have further utilized VI in CL, such as VCL meta-learning (Zhang et al. (2021)) and uncertainty-VCL (Ahn et al. (2019)). These VCL-based methods are often categorized as regularization-based approaches, as they employ Kullback-Leibler (KL) divergence as a regularization term within the variational lower bound.

Although these methods have proven to be highly effective in various CL benchmarks, it is worth noting that many of the VCL approaches primarily utilize multi-head neural networks (Nguyen et al. (2018), Ahn et al. (2019), Wang et al. (2021)). As a result, these methods are mainly evaluated in task-incremental (task-IL) scenario. In task-IL scenario, the model is provided with the task ID during testing, enabling it to focus solely on classification within the given task. However, In most cases, a model is not aware of the specific task it is facing. This scenario is known as the class-incremental (class-IL) scenario (van de Ven et al. (2022)). Class-IL scenario requires the network to infer the right class label among all those seen (Buzzega et al. (2020)), which presents a greater

level of difficulty compared to the task-IL scenario. Hence, the use of a multi-head neural network is unfeasible any more.

Few studies explore the performance of VCL in class-IL scenario. A recent study (Kessler et al. (2021)) has demonstrated that VCL is insufficient in handling continual learning in the class-IL scenario, in which the researchers attempted to replicate VCL by using a single-head neural network yet found that it resulted in severe catastrophic forgetting. More importantly, this study suggests that the superiority of multi-head VCL over other methods is not due to the propagation of the posterior, but rather because of the task-specific parameters in each head. In other words, the propagation of the posterior and the constraints of the posterior distribution on old and new tasks are not adequate for preventing forgetting in the class-IL scenario. Despite the important conclusion, this study only subject to simple experimental analysis on custom datasets without theoretical analysis. As a result, the exact reasons why VCL underperforms in class-IL scenario remain unclear.

As it is known to all, the human brain exhibits a strong advantage in addressing catastrophic forgetting, whether it is in task-IL or class-IL scenarios (Nguyen et al. (2018)), and this advantage is related to the memory consolidation mechanisms during sleep (Tadros et al. (2022)). Multiple studies in neuroscience suggest that there exist consolidation mechanisms during sleep that involve the reactivation of memory encoding from daytime learning (Buhry et al. (2011), Schreiner et al. (2021)). This reactivation process allows for the brain to review and reinforce the learned knowledge. Additionally, during the process of relearning, memories undergo distribution integration and generalization, enabling the extraction of more representative feature distributions (Klinzing et al. (2019)). This mechanism is the neural foundation for achieving CL of the brain, and hence provides valuable insights for designing brain-inspired online variational inference continual learning methods.

In this study, to clarify why VCL fails in the class-IL scenario, we first conduct a comprehensive theoretical analysis of VCL in the class-IL scenario. By decomposing CL methods in class-IL scenario into feature extractor and feature classifier components, we demonstrate that effective mitigation of catastrophic forgetting can only be achieved when the parameters of both the extractor and classifier remain as similar as possible to those used for previous tasks while learning new tasks. However, VCL fails to ensure this similarity, which no doubt results in poorer performance in the class-IL scenario. To tackle these issues, we seek a solution from the brain and propose a brain-inspired approach called **Learning within Sleeping (LwS)**. By simulating the distribution integration and generalization observed during memory consolidation in sleep, LwS achieves the concept of prior knowledge guiding posterior knowledge learning. Furthermore, LwS emulates the process of memory reactivation during sleep to impose a constraint on feature invariance. We have conducted extensive experiments on both task-IL settings and class-IL settings using a diverse range of continual learning benchmark tasks. The results show that, we proposed LwS, exhibits exceptional performance surpassing state-of-the-art methods across all these experiments.

## 2 RELATED WORKS

In this section, we first summarize the current mainstream approaches in the field of continual learning. We then focus on VCL and its related extensions. Finally, we introduce some recent brain-inspired CL methods.

**Continual Learning:** To address the issue of catastrophic forgetting, researchers have developed three types of methods. Firstly, replay-based methods consolidate memory by replaying a subset of past data stored in a buffer such as LiDER (Bonicelli et al. (2022)), ER (Rolnick et al. (2019)), A-GEM (Chaudhry et al. (2019)) and iCaRL (Rebuffi et al. (2017)). This type of method is currently the most effective approach, but it relaxes the constraint of CL by retaining old samples to some extent, allowing access to them. Secondly, parameter isolation methods assign independent parameters to each task like HAT (Serra et al. (2018)). These methods avoid accessing old samples while ensuring task isolation and optimum performance. However, they face several challenges such as an exponentially increasing parameter space with the number of tasks and poor performance in class-incremental setting. Finally, regularization-based methods penalize parameter changes related to previous tasks when learning new tasks like EWC (Kirkpatrick et al. (2017)) and SI (Zenke et al. (2017)). These methods utilize a smaller parameter space, but due to multiple tasks sharing the same network, they tend to perform less effectively in long sequential tasks. For instance, EWC

utilizes the diagonal Fisher information matrix and SI employs path integrals of gradient vector fields. Recent research has started combining these three types of methods to further enhance model performance.

**Variational Continual Learning:** The key concept in VCL is efficiently approximating the posterior distribution on models. The work introduced in (Nguyen et al. (2018)) merges online VI with Monte Carlo VI for continual learning. In (Ahn et al. (2019)), it is identified that VCL suffers from two limitations, namely increased memory cost for parameters and the absence of an effective forgetting mechanism. To address these concerns, uncertainty-regularized continual learning is proposed. Another work introduces meta-learning to VCL, leveraging dynamic mixtures at the meta-parameter level to enhance adaptability to diverse tasks. However, it is worth noting that these studies ignore the drawback that VCL can only be applied in task-IL scenario.

**Brain-inspired Continual Learning:** In recent years, there has been a growing interest among researchers in the field of CL in incorporating mechanisms from neuroscience to enhance CL methods. One such approach leverages the interaction between the hippocampus and prefrontal cortex in memory formation and proposes brain-inspired replay-based method (van de Ven et al. (2020)). This method trains a generator to simulate the brain's process of memory replay, allowing for the replay of past experiences without explicitly storing them. However, it requires extensive training of the generator and may struggle to avoid systematic errors. In (Iscen1 et al. (2020)), a remapping method is proposed, which takes advantage of the generalization process of old knowledge in the neocortex. It maps features from old tasks to the feature space of new tasks. However, this method overlooks the preservation of the feature space of the old tasks, resulting in limited effectiveness in long sequential tasks. Overall, current brain-inspired CL methods have shown some performance improvements, but they lack detailed simulation of neural mechanisms and theoretical explanations.

## 3 VARIATIONAL CONTINUAL LEARNING IN CLASS-IL SCENARIO

In this section, the reasons that VCL fails in class-IL scenario are analyzed in detail. First, via decomposing the CL algorithm in class-IL scenario into a Feature Extractor (FE) component and a Feature Classifier (FC) component, it is revealed that, in class-IL scenario, the performance of the CL algorithm improves when both the parameters of the FE and the FC remain as similar as possible during the learning process. Subsequently, with EWC and iCaRL as comparisons, the limitaions of VCL in both FE and FC are demonstrated.

### 3.1 CLASS-IL SCENARIO DECOMPOSITION

In this subsection, we present the fundamental setup of CL and discussed the formulations of task-IL scenario and class-IL scenario. CL involves training models to classify new classes in new tasks after being well-trained in old tasks. Given $T$ tasks, we partition $C$ classes into $T$ subsets $C_1, C_2, \ldots, C_T$, where $C = C_1 \cup C_2 \cup \cdots \cup C_T$ and $C_i \cap C_j = \emptyset$ for $i \neq j$. We denote $D_t = \{(x_t, y_t) | x_t \in X_t, y_t \in Y_t\}$ as the training dataset for each task $t$, where $X_t$ is a set of images and $Y_t$ represents the labels belonging to $C_t$.

There are two classical CL experiment settings named task-IL scenario and class-IL scenario. The formal expression of task-IL is:

$$\text{Training:} \quad p(\alpha_t | D_t^{train}) \qquad \text{Testing:} \quad p(D_t^{test} | \alpha_t), \quad \text{where } t \text{ is available,}$$

and class-IL is:

$$\text{Training:} \quad p(\alpha_t | D_t^{train}) \qquad \text{Testing:} \quad p(D_t^{test} | \alpha_t), \quad \text{where } t \text{ is unavailable,}$$

where $\alpha_t$ is the parameters of the model in task $t$. The expression above indicates that the class-IL scenario imposes stricter testing conditions compared to the task-IL scenario, and closer to the real world environment. Some studies (Kim et al. (2022)) divide the algorithms in a class-IL scenario into two components: within-task prediction (WP) and task-ID prediction (TP). TP is responsible for identifying the task that the model is currently performing, while WP performs class predictions within that particular task. In a word, it becomes a task-IL algorithm which the task id $t$ is predicted by another model.

In contrast, in our study, we approach class-IL algorithms from a different perspective. We split such an algorithm into two components: the Feature Extractor (FE) and Feature Classifier (FC).

Consider a class-IL model denoted as $f_{\theta,W} : \mathcal{X} \to \mathbb{R}^K$, where $K$ represents the number of classes. In this model, $\theta$ and $W$ correspond to the parameters of the network. Specifically, $\theta$ refers to the parameter of a feature extractor $h_\theta : \mathcal{X} \to \mathbb{R}^d$, where $d$ denotes the dimension of the feature space. Additionally, $W$ represents the parameters of a feature classifier $g_W : \mathbb{R}^d \to \mathbb{R}^K$. The network $f_{\theta,W}$ is represented as:

$$f_{\theta,W}(x) = g_W(h_\theta(x)), \quad x \in \mathcal{X}$$

Due to the non-i.i.d. (independent and identically distributed) nature of datasets across different tasks, the parameters $\theta$ and $W$ undergo changes as tasks arrive. For instance, after being trained on dataset $D_t$, we obtain a subset of features denoted as $Z_t$ that is extracted by the feature extractor $h_{\theta_t}$. Subsequently, based on $Z_t$, we train a feature classifier denoted as $g_{W_t}$. However, when task $t+1$ is introduced, the feature extractor is adjusted using the new dataset $D_{t+1}$ resulting in $h_{\theta_{t+1}}$, and similarly, the feature classifier becomes $g_{W_{t+1}}$. Consequently, when some previously encountered samples from $D_t$ reappear, the features extracted by $h_{\theta_{t+1}}$ denoted as $\hat{Z}_t$ no longer reside in the same feature space as $Z_t$ in most cases, and these features remain unseen by $g_{W_{t+1}}$. This phenomenon leads to catastrophic forgetting.

To address this problem, it is crucial to maintain the ability to perform feature extraction and classification for old tasks with the same proficiency as when initially learning them. This requires retaining the capability for feature extraction and classification of previous tasks even after learning new ones.

## 3.2 THE FEATURE EXTRACTOR (FE) AND FEATURE CLASSIFIER (FC)

As stated in subsection 3.1, catastrophic forgetting occurs due to the transformation of the model in a new task, resulting in changes in the feature space and feature classifier for that task. Consequently, the feature space obtained for old data using the new model is inconsistent with the feature space of new data. This inconsistency renders the new feature classifier unable to recognize the features extracted from the old samples.

An intuitive hypothesis suggests that if the new FE can effectively extract features from both old samples in the original feature space and new samples in the updated feature space, while the FC maintains its ability to classify in the original feature space and learns to classify in the new feature space, then the performance of the CL model is expected to improve.

To formally describe our hypothesis, we use cross-entropy as the performance measure of the model, the loss function of a class-IL model is denoted as:

$$\mathcal{L}_{CIL}(x) = H(y, f_\theta(g_W(x))) = -\sum_i y_i \log f_\theta(g_W(x_i)) \tag{1}$$

After learning task $t$, we obtain $\theta_t^*, W_t^* = \arg\min_{\theta_t \in \Theta, W_t \in \mathcal{W}} H(y_t, g_{W_t}(h_{\theta_t}(x_t)))$ where $(x_t, y_t) \in D_t$. Also after learning task $t+1$, we obtain $\theta_{t+1}^*, W_{t+1}^* = \arg\min_{\theta_{t+1} \in \Theta, W_{t+1} \in \mathcal{W}} H(y_{t+1}, g_{W_{t+1}}(h_{\theta_{t+1}}(x_{t+1})))$ where $(x_{t+1}, y_{t+1} \in D_{t+1})$. $D_t$ and $D_{t+1}$ are non-i.i.d.. With these notations, our hypothesis can be formally described as follows.

**Hypothesis 1**: catastrophic forgetting happens as $H(y_t, g_{W_t^*}(h_{\theta_t^*}(x_t))) \ll H(y_t, g_{W_{t+1}^*}(h_{\theta_{t+1}^*}(x_t)))$ where $(x_t, y_t) \in D_t$.

**Theorem 1**: if the condition $|\theta_{t+1}^{*'(i,j)} - \theta_t^{*(i,j)}| < \epsilon_1$, $|W_{t+1}^{*'(i,j)} - W_t^{*(i,j)}| < \epsilon_2$ and $\theta_{t+1}^{*'}, W_{t+1}^{*'} \approx \arg\min_{\theta_{t+1} \in \Theta, W_{t+1} \in \mathcal{W}} H(y_{t+1}, g_{W_{t+1}}(h_{\theta_{t+1}}(x_{t+1}))), (x_{t+1}, y_{t+1}) \in D_{t+1}$ hold, where $i$ and $j$ are indicators of matrix $\theta, W$, and $\epsilon_1, \epsilon_2$ are small positive numbers, then we have $H(y_t, g_{W_{t+1}^{*'}}(h_{\theta_{t+1}^{*'}}(x_t))) \leq H(y_t, g_{W_{t+1}^*}(h_{\theta_{t+1}^*}(x_t)))$.

**Proof:** In most cases, the objective function $H(y, f_\theta(g_W(x)))$ is continuous and differentiable in the parameter space $\Theta$ and $\mathcal{W}$. When $|\theta_{t+1}^{*'(i,j)} - \theta_t^{*(i,j)}| < \epsilon_1$ and $|W_{t+1}^{*'(i,j)} - W_t^{*(i,j)}| < \epsilon_2$, We have $H(y_t, g_{W_{t+1}^{*'}}(h_{\theta_{t+1}^{*'}}(x_t))) \approx H(y_t, g_{W_t^*}(h_{\theta_t^*}(x_t)))$. Since $H(y_t, g_{W_t^*}(h_{\theta_t^*}(x_t))) \ll H(y_t, g_{W_{t+1}^*}(h_{\theta_{t+1}^*}(x_t)))$, it follows that $H(y_t, g_{W_{t+1}^{*'}}(h_{\theta_{t+1}^{*'}}(x_t))) \leq H(y_t, g_{W_{t+1}^*}(h_{\theta_{t+1}^*}(x_t)))$.

Theorem 1 suggests that if we can maintain similarity between the parameters of FE and FC trained on task $t$ and those trained on task $t+1$ while training our model on task $t+1$, catastrophic forgetting can be mitigated.

### 3.3 VCL COMPARED WITH EWC AND iCaRL IN CLASS-IL SCENARIO

Based on Theorem 1, we further analyze why VCL fails in the class-IL scenario. To this end, we compare VCL with two classical CL algorithms that perform well in class-IL scenario, regularization-based classic CL algorithm Elastic Weight Consolidation (EWC), and replay-based classic CL algorithm Incremental Classifier and Representation Learning (iCaRL).

The classical regularization-based method EWC use the diagonal of the Fisher information matrix as a regularization term. The loss function of EWC is

$$\mathcal{L}_{EWC} = \mathcal{L}(\alpha_t) + \frac{\lambda}{2} \sum_i F^i (\alpha_t^i - \alpha_{t-1}^i)^2, F^i = \frac{1}{|D_t|} \sum_{d \in D_t} \partial \frac{\mathcal{L}(d, \alpha_{t-1}^i)^2}{\partial (\alpha_{t-1}^i)^2}, \tag{2}$$

where $\lambda$ is a importance weight for regularization term and $\alpha$ is the parameters of model including $\theta$ and $W$. As we can see, though EWC does not explicitly divide the algorithm into FE and FC phases, it effectively preserves the important parameters by constraining their changes as much as possible. The $\mathcal{L}_{EWC}$ is to find $\theta_{t+1}^{*'}, W_{t+1}^{*'} \approx \arg\min_{\theta_{t+1} \in \Theta, W_{t+1} \in \mathcal{W}} H(y_{t+1}, g_{W_{t+1}}(h_{\theta_{t+1}}(x_{t+1}))), (x_{t+1}, y_{t+1}) \in D_{t+1}$ while $\theta_{t+1}^{*'} \approx \theta_t^*$ and $W_{t+1}^{*'} \approx W_t^*$. As a result, EWC can guarantee a certain level of performance in class-IL scenario.

The classical replay-based method iCaRL retains the old parameters by knowledge distillation and prototype rehearsal. It involves storing newly encountered data in a exemplar set. Those data samples are replayed alongside the new data during training. Specifically, a distillation loss is used to reproduce the prediction scores stored in the previous step. The FE and FC are trained not only on the current task dataset $D_t$ but also on a subset of samples from the previous tasks dataset $D_{1:t-1}$. The loss function is:

$$\mathcal{L}_{iCaRL} = H(y_t, f_{\alpha_t}(x_t)) + H(f_{\alpha_{t-1}}(x_{buffer}), f_{\alpha_t}(x_{buffer})), \tag{3}$$

where $\mathcal{M}$ is a sample buffer, $(x_t, y_t) \in D_t$ and $(x_{buffer}, y_{buffer}) \in \mathcal{M}$.

Both EWC and iCaRL align with Theorem 1, hence exhibiting favorable performance in class-IL scenario.

Unfortunately, VCL fails to adhere to Theorem 1. Consider a VCL discriminative neural network model. Following an online bayesian approach, the prior distribution when learning task $t$ can be replaced by the posterior distribution after learning task $1 : t - 1$:

$$p(\alpha|D_{1:T}) \propto p(\alpha)p(D_{1:T}|\alpha) \propto p(\alpha|D_{1:T-1})p(D_T|\alpha) \tag{4}$$

In most cases the posterior distribution is intractable and approximation is required. Variational inference use a tractable and normalized distribution $q(\alpha_t) \in \mathcal{Q}$ as a approximation of $p(\alpha|D_t)$. In VCL, it employs a projection operator defined through a KL divergence minimization over the set of allowed approximation posteriors $\mathcal{Q}$:

$$q(\alpha_t) = \arg\min_{q \in \mathcal{Q}} KL \left( q(\alpha) \parallel \frac{1}{C_{t-1}} q(\alpha_{t-1}) p(D_t|\alpha) \right) \tag{5}$$

$C_t$ is the intractable normalizing constant and is not required to compute the optimum. After a series of transformations (detailed in Appendix 7.1), we obtain the loss function for VCL:

$$\mathcal{L}_{VCL} = \underbrace{KL[q(\alpha_t) \parallel q(\alpha_{t-1})]}_{(a)} - \underbrace{\mathbb{E}_{q(\alpha_t)}[\log p(D_t|\alpha_t)]}_{(b)} \tag{6}$$

In Gaussian mean-field approximation approach, term $(a)$ can be more specifically expressed as (detailed in Appendix 7.2):

$$KL[q(\alpha_t) \parallel q(\alpha_{t-1})] = \underbrace{\left[ \log \left( \frac{\sigma_1}{\sigma_2} \right) + \frac{\sigma_2^2}{\sigma_1^2} \right]}_{(1)} + \underbrace{\frac{(\mu_2 - \mu_1)^2}{2\sigma_1^2}}_{(2)} - \frac{1}{2}, \tag{7}$$

where $\alpha_{t-1} \sim \mathcal{N}(\mu_1, \sigma_1^2)$ and $\alpha_t \sim \mathcal{N}(\mu_2, \sigma_2^2)$. In essence, VCL attempts to maintain the similarity of old parameters by constraining the distribution that each parameter follows and minimize

their KL divergence. However, this approach is inadequate because even if the distributions that each parameter follows are similar, the sampled parameter instances may still differ. As a result, although these parameters perform well on new samples, they fail to capture similar features of the old samples with old parameters.

Therefore, due to only the parameter distribution constraint, VCL is unable to effectively preserve the feature extraction and feature prediction of old samples, resulting in catastrophic forgetting.

## 4 LEARNING WITHIN SLEEPING

Considering the inherent continual learning ability of the human brain, based on the neural mechanisms behind this advantage, i.e. the memory consolidation during sleep, we propose a **Learning within Sleeping (LwS)** framework to enhance the variational inference, rendering it can cope with the class-IL scenario.

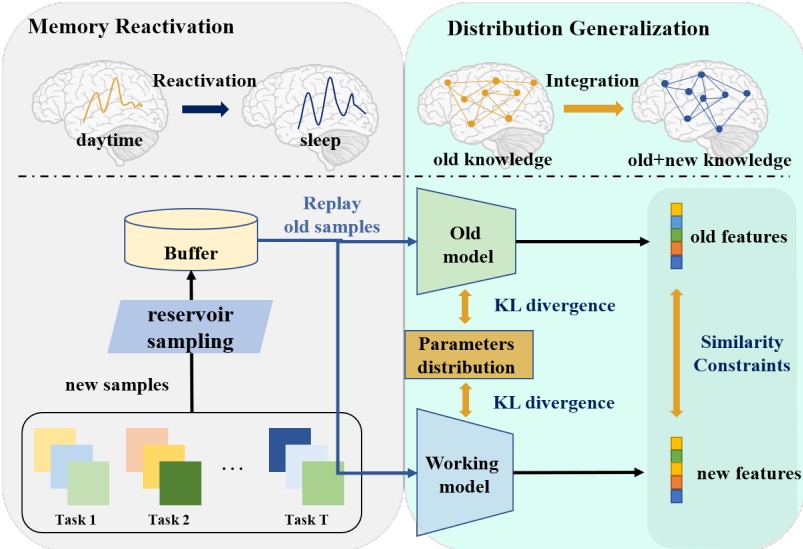

Figure 1: Memory consolidation during sleep involves two mechanisms. One is memory reactivation, which refers to the reactivation of memories encoded during the day to strengthen old knowledge, similar to the replay method in CL. The other mechanism is the integration and generalization of feature distributions during the learning process to adapt to new knowledge, corresponding to feature alignment and parameter constraints in our method.

Neuroscience studies have begun to focus on the memory consolidation mechanisms during sleep for addressing catastrophic forgetting (Brodt et al. (2023)). During sleep, previously learned experiences are reactivated, facilitating the process of memory consolidation (Buhry et al. (2011), Schreiner et al. (2021)). This reactivation is not simply a replay of the same patterns, while it involves relearning experiences, particularly the relearning of distributions (Klinzing et al. (2019)). As a result, memories obtained through new learning exhibit enhanced generalization performance. This memory consolidation mechanism during sleep bears similarities to the theoretical analysis we conducted earlier, inspiring the design of our Learning within Sleeping approach.

Inspired by the memory consolidation during sleep, we reconsider the loss function of VCL in equation (6), and introduce two main contributions:

**Memory Reactivation:** Neuronal representations are repeatedly replayed during sleep, which is crucial for memory consolidation. Taking inspiration from this, we introduce a sample buffer to store and replay samples from previous tasks alongside new data while learning new tasks. To ensure the fairness of the selected samples, we employ the reservoir sampling algorithm, which guarantees that each exemplar from the data stream has an equal probability of being selected in the buffer (Buzzega et al. (2020)). The loss function incorporating this term can be expressed as:

$$\mathcal{L}_{LwS} = H(y, f\alpha_t(x)), (x, y) \in D_t \cup \mathcal{M}. \tag{8}$$

Additionally, to improve the generalization performance of replayed samples, we propose a feature distillation term that replays features from the old parameters and constrains the features obtained from the new parameters to remain unchanged. On one hand, features possess more abstract and high-level representational power compared to samples, leading to improved model generalization. On the other hand, ensuring similarity of features for the same sample between the old and new models facilitates the ability to express old samples in the new model. Therefore, the loss function becomes:

$$\mathcal{L}_{LwS} = H(y, f\alpha_t(x)) + \mathcal{L}_{mse}(h\theta_{t-1}(x_\mathcal{M}), h_{\theta_t}(x_\mathcal{M})), (x, y) \in D_t \cup \mathcal{M}, (x_\mathcal{M}, y_\mathcal{M}) \in \mathcal{M} \quad (9)$$

**Distribution Generalization:** The theory of Bayes brain suggests that the human brain operates in a manner consistent with Bayesian inference when it comes to memory. Memories acquired during the day are consolidated and integrated during the sleep process at night, leading to the phenomenon of generalization. In VCL, the distributions of parameters serve as the memory and are integrated by preventing changes between the prior and posterior distributions. This approach aims to enable the newly learned parameter distribution to perform well not only on new tasks but also on old tasks, achieving distributional generalization. However, our analysis in section 3 suggests that constraining the distribution each parameter follows is inefficient. In this work, we attempt to constrain only the distribution of parameters in the FE after learning through point estimation. We then use this distribution as a prior to guide the new posterior distribution of parameters, treating the parameters themselves instead of the distributions they follow as memory. This process involves integrating and generalizing memory. The final loss function is as follows:

$$\mathcal{L}_{LwS} = H(y, f\alpha_t(x)) + \mathcal{L}_{mse}(h\theta_{t-1}(x_\mathcal{M}), h_{\theta_t}(x_\mathcal{M})) + KL[q(\theta_t) \parallel q(\theta_{t-1})],$$
$$(x, y) \in D_t \cup \mathcal{M}, (x_\mathcal{M}, y_\mathcal{M}) \in \mathcal{M}. \quad (10)$$

In addition, the aforementioned constraints on new and old features can actually serve as a means of distributional generalization.

---

**Algorithm 1** LwS Framework

---

**Require:** Sequence of datasets: $D = \{D_1, D_2, \ldots, D_T\}$, data buffer $\mathcal{M}$,
 1: Initialize the buffer: $\mathcal{M} \leftarrow \emptyset$
 2: run network training on task 0 with loss function:

$$\mathcal{L}_{LwS} = H(y_0, f_{\alpha_t}(x_0)), (x_0, y_0) \in D_0$$

 3: Put selected samples into buffer $\mathcal{M}$ using herding algorithm. $\mathcal{M} \leftarrow \mathcal{M} \cup \text{subset}(D_0)$
 4: **for do** $t = 1 \ldots T$
 5:     Observe the next dataset $D_t$
 6:     Update the working net posterior with loss function:

$$\mathcal{L}_{LwS} = H(y, f\alpha_t(x)) + \mathcal{L}_{mse}(h\theta_{t-1}(x_\mathcal{M}), h_{\theta_t}(x_\mathcal{M})) + KL[q(\theta_t) \parallel q(\theta_{t-1})],$$
$$(x, y) \in D_t \cup \mathcal{M}, (x_\mathcal{M}, y_\mathcal{M}) \in \mathcal{M}.$$

 7:     Update buffer $\mathcal{M}$ using reservoir sampling algorithm. $\mathcal{M} \leftarrow \mathcal{M} \cup \text{subset}(D_t)$

---

Inspired by the memory consolidation mechanisms during sleep, we propose sample replay and feature alignment constraints by simulating memory reactivation. Moreover, we propose a parameter distribution constraint method by simulating distribution generalization. These components collectively form our LwS framework. With these components, LwS satisfies the description of Theorem 1 and effectively avoids the occurrence of catastrophic forgetting in class-IL scenario.

## 5 EXPERIMENTS

### 5.1 EXPERIMENT SETUP

We test our proposal in two continual learning settings: Task-IL and Class-IL on the Mammoth framework (Pietro Buzzega (2020)), which is a CL benchmark contains many classical CL methods. When tested on the MNIST dataset, we adopted MNISTMLP model which is a two-layer fully

connected network with 100 ReLU units per layer. For the CIFAR10 and TinyImageNet datasets, we used a Resnet18 convolutional neural network. All models in this study are trained without the use of pre-training models. We use the SGD optimizer for all settings. We experiment with the following datasets: **Split MNIST**(Lecun et al. (1998)): MNIST datasets which split into 5 tasks for each task contains 2 classes. **Split CIFAR-10**(Krizhevsky (2009)): CIFAR-10 datasets which split into 5 tasks for each task contains 2 classes. **Split Tiny-ImageNet**(Le & Yang (2015)): Tiny-ImageNet datasets which split into 10 tasks for each task contains 20 classes. We evaluate the performance of LwS together with the classical bayesian method VCL, some famous regularization-based methods such as EWC, SI and some famous replay-based methods like iCaRL and A-GEM. A sample buffer is set to store samples of old tasks. We test all methods in 200, 500, 5120 three kinds of buffer size.

## 5.2 COMPARING WITH BAYESIAN METHODS ON SPLIT-MNIST DATASET

Firstly, we compare the performance of our method with that of the bayesian methods using the split-MNIST dataset. Figure 2 shows the comparison with VCL in both task-IL and class-IL on three buffer sizes. LwS provides the highest performance in all of these scenarios. In task-IL, LwS and VCL both have good performance in all tasks. But in class-IL, VCL almost forget what it learn immediately, and shows very poor accuracy in old tasks. Instead, LwS is very suitable for class-IL that provides a percentage gain of 440%, 448% and 487% over the VCL in buffer size 200, 500 and 5120. Regardless of the buffer size used, the standard deviations of the accuracy for LwS tasks were 9.70, 8.00, and 2.52, respectively. These results demonstrate that LwS exhibits strong continual learning capabilities without significant forgetting of previously learned tasks.

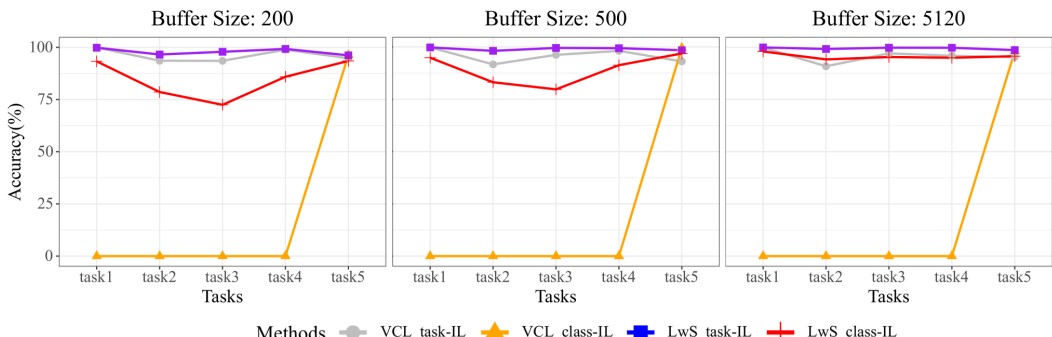

Figure 2: VCL and LwS on Split-MNIST dataset in task-IL scenario and class-IL scenario

## 5.3 COMPARING WITH NON-BAYESIAN METHODS

Compared to Bayesian methods, non-Bayesian methods tend to have higher average accuracy on old tasks. Table 1 shows the average test accuracy both in regularization-based methods like EWC, SI and replay-based methods like iCaRL and A-GEM. Since regularization-based methods do not require a sample buffer, we only conducted tests on different datasets. Usually the performance of these methods are typically not as good as replay-based methods. As for replay-based methods, we compared our method with them by testing the average accuracy on sample buffers of three different sizes: 200, 500, and 5120 samples.

The results show that our method provide considerable performance on all three datasets. In particular, our algorithm demonstrates significantly higher average accuracy than existing algorithms, especially when dealing with complex datasets like Split-CIFAR10 and Split-TinyImageNet. The performance gap between LwS and A-GEM and iCaRL shows the clear advantages of our method over both regularized-based and replay-based methods.

## 5.4 ABLATION STUDY

Sample replay has been repeatedly proven to be effective in addressing catastrophic forgetting. Therefore, to further validate the effectiveness of the two constraints we introduced, we conduct ablation experiments. We evaluat the performance improvement of our constraints on the split-MNIST

Table 1: Comparison with other works on task-IL and class-IL

| Buffer | Method | Split-MNIST | | Split-CIFAR10 | | Split-TinyImageNet | |
|---|---|---|---|---|---|---|---|
| | | Task-IL | Class-IL | Task-IL | Class-IL | Task-IL | Class-IL |
| - | oEWC | 96.95 | 21.87 | 68.29 | 19.49 | 19.20 | 7.58 |
| | SI | 95.94 | 19.87 | 68.05 | 19.48 | 36.32 | 6.58 |
| **200** | VCL | 96.21 | 18.04 | - | - | - | - |
| | iCaRL | **98.81** | 61.93 | **88.99** | **49.02** | 28.19 | 7.53 |
| | A-GEM | 98.66 | 80.11 | 83.88 | 20.04 | 22.77 | 8.07 |
| | **LwS** | 97.34 | **83.57** | 87.62 | 48.96 | **31.04** | **8.63** |
| **500** | VCL | 95.80 | 19.92 | - | - | - | - |
| | iCaRL | 98.84 | 54.87 | 88.22 | 47.55 | 31.55 | 9.38 |
| | A-GEM | 98.3 | 85.99 | **89.48** | 22.67 | 25.33 | 8.06 |
| | **LwS** | **98.95** | **87.47** | 87.12 | **48.55** | **35.91** | **10.04** |
| **5120** | VCL | 95.82 | 19.60 | - | - | - | - |
| | iCaRL | 98.82 | 58.67 | 92.23 | 55.07 | 40.83 | 14.08 |
| | A-GEM | 98.42 | 95.55 | 90.10 | 21.99 | 26.22 | 7.96 |
| | **LwS** | **99.03** | **95.62** | **94.69** | **75.69** | **58.93** | **21.24** |

*- indicates the experiments that the author were unable to run. Results of other methods referenced from (Pietro Buzzega (2020), Arani et al. (2022))*

dataset and observed positive results. We set LwS-non as LwS with only sample replya. The results demonstrat that the constraint on parameter variation named as LwS-par improved accuracy across almost all tasks, while the constraint on feature consistency named as LwS-fea exhibited even better performance enhancement in the average results.

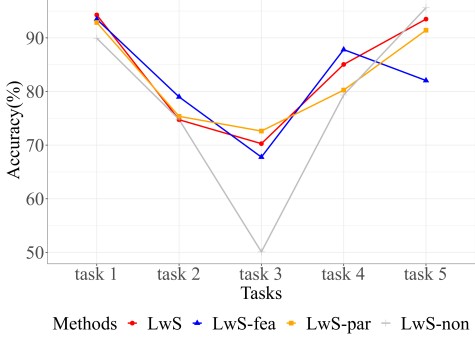

| | class-IL | task-IL |
|---|---|---|
| LwS | **83.57** | **97.34** |
| LwS-fea | 82.02 (-1.55) | 96.56 (-0.78) |
| LwS-par | 82.51 (-1.06) | 97.27 (-0.07) |
| LwS-non | 77.96 (-5.61) | 98.09 (+0.75) |

Table 2: The average accuracy on the Split-MNIST dataset after removing each component in class-IL scenario and task-IL scenario.

Figure 3: LwS ablation study in split-MNIST in buffer size 200

## 6 CONCLUSION

In this work, with comprehensive theoretical analysis and experimental results, we reveal that VCL can not solve the catastrophic forgetting on class-IL scenario, which mainly results from its inability to effectively constrain the variation of model parameters across old and new tasks. To cope with this challenge, we design a brain-inspired Learning within Sleep framework. The proposed LwS simulates memory reactivation through replaying and applies parameter constraints based on distribution generalization and feature constraints. The experiments validate that, our method achieves superior results compared with both Bayesian and non-Bayesian methods. Furthermore, we indicates the effectiveness of incorporating brain mechanisms on designing novel approaches for CL.

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

## 7 APPENDIX

### 7.1 DERIVATION OF EQ. (6)

Since we use $q(\alpha_t) \approx p(\alpha_t|D_t)$ To find $q(\alpha_t) = \arg\min_{q \in \mathcal{Q}} KL\left(q(\alpha) \parallel \frac{1}{C_{t-1}} q(\alpha_{t-1}) p(D_t|\alpha)\right)$, we consider:

$$
\begin{aligned}
\log p(D) &= \int q(\alpha) \log p(D) \, d\alpha \\
&= \int q(\alpha) \log \frac{p(D, \alpha)}{p(\alpha|D)} \, d\alpha \\
&= \int q(\alpha) \log \frac{p(D, \alpha) q(\alpha)}{p(\alpha|D) q(\alpha)} \, d\alpha \\
&= \int q(\alpha) \log \frac{p(D, \alpha)}{q(\alpha)} \, d\alpha + \int q(\alpha) \log \frac{q(\alpha)}{p(\alpha|D)} \, d\alpha \\
&= \mathcal{L}_{ELBO}(q(\alpha)) + KL[q(\alpha) \parallel p(\alpha|D)]
\end{aligned}
$$

Since $\log p(D)$ is independent of $q(\alpha)$, minimizing $KL[q(\alpha) \parallel p(\alpha|D)]$ implies maximizing $\mathcal{L}_{ELBO}(q(\alpha))$.

$$
\begin{aligned}
\mathcal{L}_{ELBO}(q(\alpha_t)) &= \int q(\alpha_t) \log \frac{p(D_t, \alpha_t)}{q(\alpha_t)} \, d\alpha \\
&= \int q(\alpha_t) \log \frac{p(D_t|\alpha_t)p(\alpha_t)}{q(\alpha_t)} \, d\alpha \\
&= \int q(\alpha_t) \log p(D_t|\alpha_t) \, d\alpha + \int q(\alpha_t) \log \frac{p(\alpha_t)}{q(\alpha_t)} \, d\alpha_t \\
&= \int q(\alpha_t) \log p(D_t|\alpha_t) \, d\alpha + \int q(\alpha_t) \log \frac{q(\alpha_{t-1})}{q(\alpha_t)} \, d\alpha \\
&= \mathbb{E}_{q(\alpha_t)}[\log p(D_t|\alpha_t)] - KL[q(\alpha_t) \parallel q(\alpha_{t-1})]
\end{aligned}
$$

Then we have:

$$
\begin{aligned}
\mathcal{L}_{VCL} &= \max \mathcal{L}_{ELBO} = \min(-\mathcal{L}_{ELBO}) \\
&= KL[q(\alpha_t) \parallel q(\alpha_{t-1})] - \mathbb{E}_{q(\alpha_t)}[\log p(D_t|\alpha_t)]
\end{aligned}
$$

## 7.2 DERIVATION OF EQ. (7)

Let $q(\theta_{t-1}) = \mathcal{N}(\mu_1, \sigma_1^2)$ and $q(\theta_t) = \mathcal{N}(\mu_2, \sigma_2^2)$, which we have:

$$
q(\theta_{t-1}) = \frac{1}{\sqrt{2\pi}\sigma_1} e^{-\frac{(\theta - \mu_1)^2}{2\sigma_1^2}}
$$

$$
\log q(\theta_{t-1}) = -\frac{1}{2} \log(2\pi) - \log(\sigma_1) - \frac{1}{2}(\frac{\theta - \mu_1}{\sigma_1})^2
$$

$$
q(\theta_t) = \frac{1}{\sqrt{2\pi}\sigma_2} e^{-\frac{(\theta - \mu_2)^2}{2\sigma_2^2}}
$$

$$
\log q(\theta_t) = -\frac{1}{2} \log(2\pi) - \log(\sigma_2) - \frac{1}{2}(\frac{\theta - \mu_2}{\sigma_2})^2
$$

So the $KL[q(\theta_t) \parallel q(\theta_{t-1})]$ can be expressed as:

$$KL[q(\theta_t) \parallel q(\theta_{t-1})]$$

$$= \int q(\theta_t) \log \frac{q(\theta_t)}{q(\theta_{t-1})} \, d\theta$$

$$= - \int q(\theta_t) \log q(\theta_{t-1}) \, d\theta + \int q(\theta_t) \log q(\theta_t) \, d\theta$$

$$= \int \frac{1}{\sqrt{2\pi}\sigma_2} e^{-\frac{(\theta-\mu_2)^2}{2\sigma_2^2}} \times \left[ \frac{1}{2}\log(2\pi) + \log(\sigma_1) + \frac{1}{2}\left(\frac{\theta-\mu_1}{\sigma_1}\right)^2 - \frac{1}{2}\log(2\pi) - \log(\sigma_2) - \frac{1}{2}\left(\frac{\theta-\mu_2}{\sigma_2}\right)^2 \right] d\theta$$

$$= \int \frac{1}{\sqrt{2\pi}\sigma_2} e^{-\frac{(\theta-\mu_2)^2}{2\sigma_2^2}} \times \left\{ \log\left(\frac{\sigma_1}{\sigma_2}\right) + \frac{1}{2}\left[ \left(\frac{\theta-\mu_1}{\sigma_1}\right)^2 - \left(\frac{\theta-\mu_2}{\sigma_2}\right)^2 \right] \right\} d\theta$$

$$= \mathbb{E}_{q(\theta_t)} \left\{ \log\left(\frac{\sigma_1}{\sigma_2}\right) + \frac{1}{2}\left[ \left(\frac{\theta-\mu_1}{\sigma_1}\right)^2 - \left(\frac{\theta-\mu_2}{\sigma_2}\right)^2 \right] \right\}$$

$$= \log\left(\frac{\sigma_1}{\sigma_2}\right) + \frac{1}{2\sigma_1^2}\mathbb{E}_{q(\theta_t)}[(\theta-\mu_1)^2] - \frac{1}{2\sigma_2^2}\mathbb{E}_{q(\theta_t)}[(\theta-\mu_2)^2]$$

$$= \log\left(\frac{\sigma_1}{\sigma_2}\right) + \frac{1}{2\sigma_1^2}\mathbb{E}_{q(\theta_t)}[(\theta-\mu_1)^2] - \frac{1}{2}$$

$$= \log\left(\frac{\sigma_1}{\sigma_2}\right) + \frac{1}{2\sigma_1^2}\mathbb{E}_{q(\theta_t)}[(\theta-\mu_2+\mu_2-\mu_1)^2] - \frac{1}{2}$$

$$= \log\left(\frac{\sigma_1}{\sigma_2}\right) + \frac{1}{2\sigma_1^2}\mathbb{E}_{q(\theta_t)}[(\theta-\mu_2)^2 + 2(\theta-\mu_2)(\mu_2-\mu_1) + (\mu_2-\mu_1)^2] - \frac{1}{2}$$

$$= \log\left(\frac{\sigma_1}{\sigma_2}\right) + \frac{1}{2\sigma_1^2}\{\mathbb{E}_{q(\theta_t)}[(\theta-\mu_2)^2] + 2(\mu_2-\mu_1)\mathbb{E}_{q(\theta_t)}[(\theta-\mu_2)] + \mathbb{E}_{q(\theta_t)}[(\mu_2-\mu_1)^2]\} - \frac{1}{2}$$

$$= \log\left(\frac{\sigma_1}{\sigma_2}\right) + \frac{1}{2\sigma_1^2}[\sigma_2^2 + 0 + (\mu_2-\mu_1)^2] - \frac{1}{2}$$

$$= \log\left(\frac{\sigma_1}{\sigma_2}\right) + \frac{\sigma_2^2 + (\mu_2-\mu_1)^2}{2\sigma_1^2} - \frac{1}{2}$$

$$= \left[\log\left(\frac{\sigma_1}{\sigma_2}\right) + \frac{\sigma_2^2}{\sigma_1^2}\right] + \frac{(\mu_2-\mu_1)^2}{2\sigma_1^2} - \frac{1}{2}$$

