# OpenReview forum: "Learning within Sleeping: A Brain-Inspired Bayesian Continual Learning Framework"
_ICLR.cc/2024/Conference — ICLR 2024 Conference Withdrawn Submission_

### Official Review · Reviewer_ET4e · 2023-10-23

**Soundness:** 2 fair
**Presentation:** 2 fair
**Contribution:** 2 fair
**Rating:** 5
**Confidence:** 4

**Summary:**

This paper introduces a method of continual learning inspired by a human brain mechanism, namely, memory consolidation during sleeping. The proposed method was compared with the original VCL and other existing methods in the literature over Split MNIST, Split CIFAR-10, and Split Tiny-ImageNet datasets. The proposed method showed superior performance, especially in the class incremental learning scenario.

**Strengths:**

The introduction and the relation works sections are well written and clearly describe the goal of the work.

The inspiration from a human brain mechanism is exciting, and the proposed learning within sleeping (LwS) showed remarkable performance improvements.

**Weaknesses:**

While the idea is interesting and meaningful, the proposed method seems to combine the existing methods in the literature rather than a new one. In learning while sleeping, the memory reactivation is similar to the replay-based methods, and the distribution generalization is similar to the regularization-based methods.

From a neuroscience perspective to this reviewer’s knowledge, learning during sleep is more related to the stimulus given before sleeping (memory reactivation); thus, it has nothing to do with a new stimulation (no sample from a new task is given). In this regard, this reviewer believes that the proposed LwS doesn’t follow the neural mechanism.

Many typos in the context and the equations make it challenging to read and understand the paper. Especially for the equations, please carefully check the super/subscripts.

**Questions:**

Check the Weaknesses above.

For the consistency, the notations in Eq. (1) should be corrected as follows: (left) $f_{\theta}(g_{W}(x))$ -> $f_{\theta, W}(x)$ or $g_{W}(h_{\theta}(x))$; (right) $f_{\theta}(g_{W}(x_{i}))$ -> $f_{\theta, W}(x_{i})$ or $g_{W}(h_{\theta}(x_{i}))$.

Since the notations are unclear, it isn't very clear for Hypothesis 1, Theorem 1, and the Proof. How are the notations of $(\theta_{t+1}^{*’}, W_{t+1}^{*’})$ and $(\theta_{t+1}^{*}, W_{t+1}^{*})$ different? Especially the last sentence in Proof needs checking carefully, “~, it follows that $H~~~ <= H~~~~$.”

In Algorithm 1, $D_{0}$ is not defined.

The authors also need to provide the results of VCL and LwS on other datasets in task-IL and class-IL scenarios.

In Figure 2, the markers in the graphs and the legends do not match.

---

> ### Author Response · Authors · 2023-11-20
>
> Thanks for your constructive comments and suggestions. We have carefully considered the feedback provided and decide to withdraw this manuscript.
>
> We thank the reviewer for recognizing and acknowledging the novelty of our idea.
>
> Although each of our modifications aligns with the theoretical derivations and sleep mechanisms described in the paper, and differs in implementation from existing parameter regularization and feature distillation methods, we acknowledge that the innovativeness of our algorithmic changes may not be sufficiently apparent. There are two main reasons for this: first, integrating brain mechanisms into algorithmic improvements is a necessary but challenging task, so further exploration is required to establish a reasonable correspondence between brain mechanisms and specific algorithm designs. Second, the VCL algorithm itself is based on parameter regularization, which necessitates that our improvements build upon this foundation, making our method appear less novel. Therefore, in our future work, we will delve deeper into incorporating relevant brain mechanisms for more profound algorithmic enhancements to achieve superior performance.
>
> Regarding the second question you raised, it depends on the understanding of what the stimulus given before sleeping are. For instance, in a completely blank brain, after receiving stimuli on the first day, what is consolidated during the first night of sleep is solely the stimuli received on that first day. However, when sleeping on the night of the second day, the brain integrates stimuli from both the first and second days, rather than just the stimuli received on the second day (even though the proportion of stimuli received on the second day is higher). In our algorithm, we simulate this integration process by aligning parameters and features. Therefore, in our view, LwS follows the brain mechanisms.
>
> If you are interested in this brain-inspired algorithm, we would be more than willing to discuss it with you after it becomes open for discussion.
>
> We sincerely appreciate the invaluable feedback again, and we will make further modifications to the work based on these suggestions.

---

### Official Review · Reviewer_LsLM · 2023-10-29

**Soundness:** 2 fair
**Presentation:** 1 poor
**Contribution:** 1 poor
**Rating:** 3
**Confidence:** 4

**Summary:**

The paper investigates the reason why variational continual learning does not work well for the class-incremental learning and then proposes two techniques to improve the performance including replay and knowledge distillation. The resulting method indeed shows enhanced performance on the class-incremental learning scenario from the results demonstrated in the paper.

**Strengths:**

The mathematical formulation seems to be correct.
The effectiveness of replay and feature distillation are again demonstrated.

**Weaknesses:**

Motivation of this paper is not clear. The paper extends on variational continual learning to the class-incremental learning scenario but it is unclear what are the particular benefits of extending over VCL rather than other approaches.

It is nice that the paper writes the problem and solution in a clear mathematical form but what the conclusions and solutions can be easily explained with natural language. The mathematical proof seems unnecessary, which can be moved to supplemental.

The novelty of the proposed method is very limited. The resulting solution is replay (similar to all replay methods like GEM, DER, DER++, GSS, [1], [2], [3], etc.) and perform distillation (similar to LwF) on the feature extraction part. But the solution is widely recognized as useful and proved in many prior works. I cannot identify any other contributions.

The comparison is not convincing. Results are not compared to more recent works such as DER and [1][2][3] so it is not convincing that the result is significant at the current time.

[1] Rishabh Tiwari et al. GCR: Gradient Coreset based Replay Buffer Selection for Continual Learning. CVPR 2022
[2] Elahe Arani et al. Learning Fast, Learning Slow: A General Continual Learning Method based on Complementary Learning System. ICLR 2022
[3] Da-Wei Zhou et al. A Model or 603 Exemplars: Towards Memory-Efficient Class-Incremental Learning. ICLR 2023

**Questions:**

No questions.

---

> ### Author Response · Authors · 2023-11-20
>
> Thanks for your constructive comments and suggestions. We have carefully considered the feedback provided and decide to withdraw this manuscript. I would to address some of the questions you raised.
>
> The innovative aspects of our work can be summarized as follows:
>
> 1. Theoretical analysis for the first time identifies the reasons behind the poor performance of the VCL algorithm in class-incremental learning scenarios.
>
> 2. Comprehensive demonstration of the impact of new data on catastrophic forgetting, providing insights for the development of new continual learning algorithms to overcome catastrophic forgetting.
>
> 3. Introduction, for the first time, of the brain mechanism of sleep memory consolidation to develop novel continual learning algorithms.
>
> The aim of this work is to identify the reasons why VCL, a classic and promising method, performs poorly in Class-IL scenarios and to propose a brain-inspired algorithm to address this issue. One oversight on our part was neglecting to provide an explanation for why improvements were made to the VCL method. In fact, this stems from the understanding in numerous neuroscience studies that the human brain operates based on Bayesian inference, and VCL is designed around Bayesian inference. Therefore, among the various algorithms emulating the continuous learning ability of the human brain, VCL is a particularly promising method.
>
> Your suggestions regarding the theoretical aspects are highly constructive, and we appreciate this feedback. The theoretical section aims to elucidate the reasons behind the failure of the VCL method, leading to an excessive amount of background information. We plan to rewrite and include this information in the supplementary materials.
>
> Thank you for your additions concerning comparative algorithms and relevant research literature; we will incorporate additional experiments accordingly.
>
> Once again, we appreciate your insights, and we will make further modifications to the work based on this feedback.

---

### Official Review · Reviewer_KMAy · 2023-10-30

**Soundness:** 2 fair
**Presentation:** 2 fair
**Contribution:** 3 good
**Rating:** 5
**Confidence:** 4

**Summary:**

This paper proposed a new Bayesian continual learning method inspired by human memory consolidation process during sleep. The idea is interesting and novel, but in implementation it seems the method still trying to find a balance between the old and new data/features on a unified model, which human brain may not necessarily working in this way to process new memory. Results indicate this method is effective in a range of benchmarking experiments but fails to outperform some existing algorithms.

**Strengths:**

1. The idea of memory consolidation is novel and it is interesting to dig this further. A strong and convincing point of the justification in the paper is that model transformation by new data is a primary reason that cause the forgetting issue.

2. Split the model into FE and FC seems a good idea to make this complicated problem into simpler and easier ones.

3. The way of presentation is good, easy to follow.

**Weaknesses:**

1. It seems this work was somehow rushed, especially the results section. I found a number of obvious typos, mainly spelling error. It would be good to at least carefully proofread few times before submission.

2. The idea and the inspiring source is novel, but unfortunately when it comes to implementation, the proposed method is still trying to find a balance between old and new data/features. I doubt this might be also the reason that makes its performance fairly good but not outstanding amongst existing work.

**Questions:**

1. Fig 1, top right, the yellow connected points before integration, shouldn't it named 'new knowledge' rather than 'old knowledge'?

2. It seems a higher buffer number favours the proposed work, could the authors further justify this?

---

> ### Author Response · Authors · 2023-11-20
>
> Thank you very much for recognizing and acknowledging our idea. This is an innovative aspect that we are highly confident in and will continue to explore.
>
> We greatly appreciate your concerns regarding the methodological aspects. Our approach is based on an understanding of memory consolidation during the sleep process, specifically mapping encoded old memories to new memory spaces. This represents a compromise between brain mechanisms and deep learning methods. While it may not align perfectly with brain mechanisms, the necessity for algorithmic improvements arises from the spatial changes that lead to catastrophic forgetting in deep learning methods.
>
> Due to time constraints, further experimental validation has not been extensively covered, and we plan to enhance the reliability of this work in subsequent research. Your point about the advantageous impact of a larger buffer size on our method is intriguing, and we are keen on further exploring this aspect.
>
> If you find our work interesting, we would be more than willing to communicate with you after the anonymity period concludes.

---

### Official Review · Reviewer_HYLs · 2023-11-01

**Soundness:** 2 fair
**Presentation:** 2 fair
**Contribution:** 3 good
**Rating:** 5
**Confidence:** 4

**Summary:**

In this paper, the authors propose a novel brain-inspired Learning within Sleeping (LwS) approach to solve continuous learning problems, which simulates memory reactivation and distribution generalization mechanisms. It can be considered as a combination of a rehearsal technique with parameter distribution regularization. The experiments conducted show the performance of LwS on various CL benchmark scenarios, including both task incremental and class incremental learning.

**Strengths:**

(1) The paper is clearly written and easily understood.

(2) The proposed solution originates from a theoretical analysis of the problem of catastrophic forgetting in VCL models.

(3) The experimental results demonstrate the performance of LwS compared to the state-of-the-art, especially in class-IL scenarios.

**Weaknesses:**

(1) Although somewhat original, the model presented seems to be a combination of known solutions.

(2) The authors claim to "conduct a comprehensive theoretical analysis of VCL in the class-IL scenario". I find this a strong statement, so one would expect a serious in-depth analysis. Meanwhile, the reasoning presented in Section 3 seems only to formalize (perhaps sometimes in an unnecessarily complicated way) some natural observations.

(3) Since LwS includes a data buffer, its fair experimental competitors are the other replay-based models. In this respect, the authors only present a comparison with iCaRL (2017) and A-GEM (2019).

**Questions:**

(1) Regarding Tab. 1, what is the role of a buffer in the VCL? (Note that increasing a buffer does not increase accuracy).

(2) I would suggest expanding the experimental setup to include newer rehearsal-based competitors.

(3) Will you share the source code for performed experiments?

(4) Minor comments:

p. 7, Eqs. (9) and (10): $h\theta_{t-1} \to h_{\theta_{t-1}}$,

p. 8, l. 1 from bottom: evaluate $\to$ evaluate,

p. 9, under Tab. 1: replya $\to$ reply,

p. 9, l. 3-2 from bottom: that, our $\to$ that our, indicates $\to$ indicate.

---

> ### Author Response · Authors · 2023-11-20
>
> Thank you for the feedback on our work; your comments are highly valuable for my future improvements.
>
> Regarding originality, our primary innovation lies in the introduction of brain mechanisms to address the shortcomings of the VCL method in class-incremental learning. While we acknowledge the need for further enhancement in the originality of the method itself.
>
> The suggestion about the slight redundancy in the theoretical analysis is constructive. We are considering rewriting this section to provide a more robust and comprehensive theoretical analysis, demonstrating the VCL's shortcomings.
>
> Although we compared our method with existing replay methods, we acknowledge the lack of replication for the latest replay-related works due to time constraints. We plan to reproduce these works in subsequent research and incorporate the results into the comparative experiments.
>
> Once these tasks are completed, we intend to open-source our code.
>
> If you find our work intriguing, we would be more than willing to discuss it with you after the anonymity period concludes.